# LncRNA *LINC00518* Acts as an Oncogene in Uveal Melanoma by Regulating an RNA-Based Network

**DOI:** 10.3390/cancers12123867

**Published:** 2020-12-21

**Authors:** Cristina Barbagallo, Rosario Caltabiano, Giuseppe Broggi, Andrea Russo, Lidia Puzzo, Teresio Avitabile, Antonio Longo, Michele Reibaldi, Davide Barbagallo, Cinzia Di Pietro, Michele Purrello, Marco Ragusa

**Affiliations:** 1Department of Biomedical and Biotechnological Sciences—Section of Biology and Genetics, University of Catania, 95123 Catania, Italy; barbagallocristina@unict.it (C.B.); dbarbaga@unict.it (D.B.); dipietro@unict.it (C.D.P.); purrello@unict.it (M.P.); 2Department of Medical, Surgical Sciences and Advanced Technologies G.F, Ingrassia—Section of Anatomic Pathology, University of Catania, 95123 Catania, Italy; rosario.caltabiano@unict.it (R.C.); giuseppe.broggi@gmail.com (G.B.); lipuzzo@unict.it (L.P.); 3Eye Clinic, University of Catania, 95123 Catania, Italy; andrearusso2000@hotmail.com (A.R.); t.avitabile@unict.it (T.A.); antlongo@unict.it (A.L.); mreibaldi@libero.it (M.R.)

**Keywords:** UM, lncRNAs, miRNA sponge, miRNA protector, MITF, ceRNA network, RNA–RNA interactions, migration, EMT, hypoxia

## Abstract

**Simple Summary:**

Uveal melanoma (UM) is the most frequent primary tumor of the eye in adults. Although molecular alterations on protein-coding genes have been associated with the development of UM, the role of non-coding RNAs and their competitive endogenous networks remain poorly investigated. Starting from a computational analysis on UM expression dataset deposited in The Cancer Genome Atlas, we identified the long non-coding RNA *LINC00518* as a potential oncogene. We then experimentally evaluated *LINC00518* and its supposed RNA signaling in human biopsies and in vitro functional assays. The results obtained suggest that *LINC00518*, under potential transcriptional control by *MITF*, regulates an RNA–RNA network promoting cancer-related processes (i.e., cell proliferation and migration). These findings open the way to the characterization of the unknown RNA signaling associated with UM and pave the way to the exploitation of a potential target for RNA-based therapeutics.

**Abstract:**

Uveal melanoma (UM) is the most common primary intraocular malignant tumor in adults; little is known about the contribution of non-coding RNAs (ncRNAs) to UM pathogenesis. Competitive endogenous RNA (ceRNA) networks based on RNA–RNA interactions regulate physiological and pathological processes. Through a combined approach of in silico and experimental biology, we investigated the expression of a set of long non-coding RNAs (lncRNAs) in patient biopsies, identifying *LINC00518* as a potential oncogene in UM. The detection of *LINC00518* dysregulation associated with several in vitro functional assays allowed us to investigate its ceRNA regulatory network and shed light on its potential involvement in cancer-related processes, such as epithelial to mesenchymal transition (EMT) and CoCl_2_-induced hypoxia-like response. In vitro transient silencing of *LINC00518* impaired cell proliferation and migration, and affected mRNA expression of *LINGO2*, *NFIA*, *OTUD7B*, *SEC22C*, and *VAMP3*. A “miRNA sponge” and “miRNA protector” model have been hypothesized for *LINC00518*-induced regulation of mRNAs. In vitro inhibition of *MITF* suggested its role as a potential activator of *LINC00518* expression. Comprehensively, *LINC00518* may be considered a new oncogene in UM and a potential target for RNA-based therapeutic approaches.

## 1. Introduction

Uveal melanoma (UM) is the most common primary intraocular malignant tumor in adults. UM originates from melanocytes residing in both the anterior and posterior region of the uvea, namely, iris and choroid and ciliary body, respectively; the choroid represents the most frequent site of origin of the tumor (about 85%), followed by the ciliary body (5–8%) and the iris (3–5%) [1,2]. UM shows a poor patient survival and a high frequency of metastases, which occur in about 50% of patients with a strong tropism to the liver [3]. To date, chromosome alterations (including loss of 1p, 3, 6q, 8p, and gain of 1q, 6p, 8q) and mutations of specific genes have been reported to be associated with UM etiopathogenesis; however, its molecular bases are still under investigation. UM patients frequently bear mutually exclusive mutations in the GTPases *GNAQ* (G protein subunit alpha q) and *GNA11* (G protein subunit alpha 11), causing their constitutive activation. Consequently, downstream pathways associated with cell proliferation, such as mitogen-activated protein kinase (MAPK) pathway, phosphatidylinositol 3-kinase (*PI3K*)/Akt pathways, and Yes-activated protein (*YAP*) pathway, are also constitutively activated [1]. *BAP1* (BRCA1 associated protein 1) mutations are also very frequent, occurring in about 47% of patients, and are associated with increased risk of metastasis. The most of *BAP1* mutations lead to the absence of the functional protein, as well as the loss of chromosome 3 may result in *BAP1* monosomy, thus causing reduced levels of the *BAP1* protein [4,5], suggesting a tumor suppressive role for this gene in UM. Despite this evidence on coding RNAs, little is known about the contribution of non-coding RNAs (ncRNAs) to UM pathogenesis. Growing evidence over the last two decades clearly shows that ncRNAs play a fundamental role in tumor onset, progression and dissemination [6]. The ncRNA class is mainly constituted by microRNAs (miRNAs) and long non-coding RNAs (lncRNAs), which control the expression of protein-coding genes in different ways. It is interesting to note that miRNAs can bind mRNAs, and lncRNAs can sponge miRNAs. Therefore, when miRNAs bind their mRNA targets, mRNA stability decreases. In contrast, when lncRNAs sequester miRNAs, the degradation and/or inhibition of translation of mRNA targets are reduced. These RNA–RNA interactions establish a competitive endogenous RNA (ceRNA) regulatory network responsible for the regulation of physiological and pathological processes [7]. Many studies on ncRNAs have been performed in cancer, including UM. We have previously reported the altered expression of miRNAs in tumor tissues, body fluids, and exosomes from UM patients [8]. LncRNAs have recently been associated with several cancer models [9,10,11,12] and also UM carcinogenesis [13,14]; in particular, a recent study showed that the oncogenic function of *MALAT1* (metastasis associated lung adenocarcinoma transcript 1) in UM is performed through the regulation of an RNA-based network [15]. Given their strong involvement in carcinogenesis, lncRNAs may be considered as new potential targets for innovative RNA-based therapeutic approaches [12]. Compared to proteins, RNAs are more “druggable” because their targeting is mainly based on sequence complementarity. Such features make these molecules new targets for RNA-based drugs easy to design and inexpensive to synthetize [16]. The aim of this study was to investigate the RNA network underlying UM pathogenesis. Through a combined approach of in silico and experimental biology, we investigated the expression of a set of lncRNAs in patient biopsies and identified the lncRNA *LINC00518* as a potential oncogene in UM. The detection of *LINC00518* dysregulation associated with several in vitro functional assays allowed us to investigate its ceRNA regulatory network and shed light on its potential involvement in cancer-related processes.

## 2. Results

### 2.1. Selection of lncRNAs from the UM TCGA Dataset

Selection of lncRNAs potentially associated with UM was performed by using expression data available in the UM dataset of The Cancer Genome Atlas (TCGA). We used *MET* (MET proto-oncogene, receptor tyrosine kinase) and *BAP1* as “decoy” genes because of their previously reported altered expression (upregulation and downregulation, respectively) in UM, aiming to identify lncRNAs acting as potential oncogenes (positive correlation of expression with *MET* and negative with *BAP1*) or potential tumor suppressor genes (negative correlation with *MET* and positive with *BAP1*). Accordingly, we selected 11 lncRNAs: *AGAP2-AS1*, *CD27-AS1*, *CECR5-AS1*, *LINC00152* (*CYTOR*), *LINC00240*, *LINC00518*, *LINC00634*, *LINC01278*, *SLCO4A1-AS1*, *TPT1-AS1*, and *TTC28-AS1*.

### 2.2. Differential Expression of lncRNAs in Patient Biopsies

Expression of the selected lncRNAs was investigated in slices of formalin-fixed paraffin-embedded (FFPE) tumor and normal adjacent tissue obtained from eyes enucleated from 41 UM patients. Real-Time PCR reactions showed that two out of the eleven tested lncRNAs, namely, *LINC00518* and *LINC00634*, were significantly overexpressed in UM tissues compared to normal ones (*LINC00518*: average fold change = 33.3; *p*-value < 0.0001; *LINC00634*: average fold change = 2.1; *p*-value = 0.005) (Figure 1).

Given its strong upregulation in tumor tissue compared to normal tissue, we focused our attention on *LINC00518*. First, we investigated *LINC00518* expression in vitreous humor, serum, and exosomes from both vitreous and serum obtained from UM patients and unaffected individuals. PCR results showed no detectable levels of *LINC00518* in any of the sample types (data not shown). Moreover, we analyzed in tissue biopsies the expression of *MET* and *BAP1*, the two “decoy” genes used to select lncRNAs in the UM TCGA dataset. We confirmed the upregulation of *MET* (fold change = 4.79, *p*-value = 0.01) and the downregulation of *BAP1* (fold change = −1.33, *p*-value = 0.01). We also evaluated the correlation of expression between *LINC00518* and *MET* or *BAP1*. In the entire cohort, including both tumor and normal tissues, *LINC00518* showed a significant positive correlation with *MET* (r-value = 0.53, *p*-value ≤ 0.0001) and a not significant positive correlation with *BAP1* (r-value = 0.24, *p*-value = 0.06). The same analysis was performed in a sub-cohort including only tumor tissues; the positive correlation trends were confirmed, but the *p*-values were not significant (*LINC00518* vs. *MET*: r-value = 0.31, *p*-value = 0.12, *LINC00518* vs. *BAP1*: r-value = 0.35, *p*-value = 0.06).

We investigated whether *LINC00518* expression correlated with clinicopathological parameters of the study participants. No significant correlation was observed between *LINC00518* expression and age (*p*-value = 0.56), tumor diameter (*p*-value = 0.09), and tumor thickness (*p*-value = 0.2). Moreover, no significant difference in *LINC00518* expression was found in patients regarding sex (*p*-value = 0.31), tumor cell type (*p*-value = 0.16), or pathological stage (*p*-value = 0.84).

### 2.3. LINC00518 May Be Involved in Metastasis-Related Processes Rather Than Cell Proliferation

The involvement of *LINC00518* in cell proliferation was investigated through two different approaches. Modulation of *LINC00518* induced by cell cycle activation was assessed after serum starvation at both early and delayed time points; a delayed response of *LINC00518* to serum deprivation was also investigated. Serum supplement only showed a slight downregulation of *LINC00518* at 30 min, while serum response factor (*SRF*) mRNA increased its expression at each time point confirming the efficiency of the treatment (Figure 2A). Moreover, no significant alteration of *LINC00518* in response to serum-induced activation or arrest of cell cycle was observed at 24 h after serum supplement/deprivation (Figure 2B).

*LINC00518* expression was evaluated after U0126-induced inhibition of the MAPK pathway, a key process in cell proliferation and survival, previously associated with UM pathogenesis. To assess the most effective concentration of the inhibitor, cells were treated with 12.5 µM and 25 µM U0126 and viability was investigated at different time points. The MTT assay showed that the 25 µM concentration reduced cell viability at all investigated time points, while the 12.5 µM concentration induced a decrease only at 24 h post-treatment (PT) (Figure 2C). Based on these results, expression analysis was performed after treatment with 25 µM U0126. Similar to serum starvation results, *LINC00518* did not show an altered expression at any time point (Figure 2D), supporting the hypothesis that its levels are not influenced by cell cycle modulation.

On the contrary, *LINC00518* showed an altered expression in interleukin 6 (IL6)-induced epithelial–mesenchymal transition (EMT) in two UM cell lines, 92.1 and MEL270. After IL6 treatment, we investigated the expression of the mRNAs of EMT markers, showing a decreased expression of *CDH1* (E-cadherin) and increased levels of *ZEB2* (zinc finger E-box binding homeobox 2) at 48 h PT, while *VIM* (vimentin) levels raised at both 24 and 48 h PT in 92.1 cells. These data confirmed the efficiency of IL6 treatment in inducing EMT. *LINC00518* showed significantly increased expression in treated cells at both 24 and 48 h PT (Figure 3, Appendix A). In MEL270 cells, IL6 treatment induced increased levels of *VIM* at 48 h and *ZEB2* at both 24 and 48 h PT. *LINC00518* expression showed a significant increment at both 24 and 48 h PT, even if fold changes were considerably smaller than those observed in 92.1 cells (Figure 3, Appendix A). EMT was also triggered by treating UM cells with hepatocyte growth factor (HGF), which induced the alteration of only one marker in 92.1 cells, namely the upregulation of *VIM* at 48 h PT. Similarly, *LINC00518* expression was increased at 48 h PT with HGF, even if with a lower fold change compared to IL6 treatment (Figure 3, Appendix A). Unfortunately, HGF failed to induce EMT in MEL270 cells, despite the increased concentration used to treat the cells (Figure 3, Appendix A).

The involvement of *LINC00518* in the activation of *HIF1A* (hypoxia inducible factor 1 subunit alpha), a key event in angiogenetic processes, was investigated by treating UM cells with CoCl_2_ (cobalt chloride). The efficiency of CoCl_2_ treatment was assessed by analyzing mRNA levels of *VEGFA* (vascular endothelial growth factor A), targeted by *HIF1A*, which showed strongly increased levels at both 24 and 48 h PT in 92.1 cells. *LINC00518* expression was also induced by CoCl_2_ at 48 h PT (Figure 4, Appendix A). Similar effects were observed in MEL270 cells, were *VEGFA* expression increased at both 24 and 48 h PT, while *LINC00518* levels raised at 24 h PT (Figure 4, Appendix A).

Taken together, these results would suggest an involvement of *LINC00518* in metastasis-related pathways, while its expression may not be controlled by cell cycle progression.

### 2.4. LINC00518 Transient Silencing Affects the RNA-Based Network

To deeply investigate the function of *LINC00518* in UM cells, transient silencing of the lncRNA was achieved by siRNA transfection. Transfection efficiency was evaluated in Real-Time PCR, showing in all replicates an efficiency >75%. Alteration of cell viability after *LINC00518* silencing was investigated by the MTT assay, showing a significant, even though slight reduction at both 24 and 48 h post-transfection in both 92.1 and MEL270 cells (Figure 5A,B). *LINC00518* transient silencing also significantly impaired cell migration, especially in MEL270 cells (Figure 5C,D).

### 2.5. RNA Molecular Axes of LINC00518

The increased expression of *LINC00518* observed in UM tissue suggested a putative oncogenic role for this lncRNA. To investigate the molecular function of this lncRNA, we investigated the subcellular localization of *LINC00518*. We isolated RNA from the nuclear and cytoplasmic fractions and analyzed *LINC00518* in both, together with *MALAT1*, predominantly nuclear, and *CDR1-AS* (CDR1 antisense RNA), preferentially located in the cytoplasm. Our data showed that *LINC00518* is predominantly cytoplasmic in both 92.1 and MEL270 cells (Figure 6).

Given its preferential cytoplasmic localization, we hypothesized a ceRNA network centered on the “miRNA sponge” role of *LINC00518*. We then computationally identified the potential RNA interactors of *LINC00518*. As described in Methods, based on correlation of expression and sequence complementarity, we selected a set of miRNAs potentially sponged by *LINC00518*, and a set of mRNAs, targeted by these miRNAs, whose expression was positively correlated with the lncRNA. Appendix A and Appendix A show the list of selected miRNAs and mRNAs, respectively. Interestingly, in agreement with our hypothesis, the selected miRNAs have been previously reported to act as tumor suppressors in several cancer models, except for miR-3191-3p, for which no role in cancer has been reported to date (Appendix A).

The expression of the six miRNAs selected as potentially sponged by *LINC00518* was evaluated at 24 and 48 h after transfection, but no significant alteration was observed in 92.1 and MEL270 cells. Correlation of expression between lncRNA and each miRNA was also calculated: a negative trend was observed for all miRNAs, except for miR-3191-3p, in 92.1 cells, and for miR-143-3p and miR-199a-5p in MEL270 cells (Table 1). Expression analysis was also performed for the mRNAs targeted by the analyzed miRNAs. In 92.1 cells, three out of fifteen targets, namely *OTUD7B*, *SEC22C*, and *VAMP3*, showed a significant decrease of expression at 24 h post-transfection. Moreover, a significant positive correlation of expression with *LINC00518* was observed for *LINGO2*, *NFIA* and *VAMP3* (Table 1). In MEL270 cells, *KLF8*, *LINGO2*, *NFIA*, and *OTUD7B* showed significantly reduced expression at 24 h post-transfection, *SEMA6A* at 48 h post-transfection and *SEC22C* at both time points (Table 1). A significant positive correlation of expression with *LINC00518* was observed for *AHCYL2*, *CRTAP*, *ENTPD1*, *F11R*, *IFFO2*, *IP6K1*, *OTUD7B*, *RAB43*, *SEC22C*, and *VAMP3* (Table 1). Overall, *LINGO2*, *NFIA*, *OTUD7B*, *SEC22C*, and *VAMP3* downregulation and/or positive expression correlation with the lncRNA were reported for both cell lines after *LINC00158* silencing, suggesting that the expression of these mRNAs may be indirectly regulated by *LINC00518*.

For mRNAs regulated by *LINC00518* silencing, expression and correlation analyses were also performed on UM biopsies. These analyses confirmed the increased levels of these mRNAs in tumor compared to normal adjacent tissue and the significant positive correlation of expression with *LINC00518* (Figure 7).

The expression of these mRNAs was also evaluated in UM cells treated with EMT-inducers, IL6 and HGF, to evaluate whether their alterations were consistent with those observed for *LINC00518*. In 92.1 cells, three of the five mRNA targets showed an altered expression after IL6 treatment: in particular, the expression of *NFIA* and *VAMP3* was increased at both 24 and 48 h PT, while *SEC22C* was upregulated at 48 h PT. All of the analyzed mRNAs, except for *SEC22C*, showed a significant positive expression correlation with *LINC00518* (Figure 8, Appendix A). Similarly, HGF induced an increased expression of *NFIA* at both 24 and 48 h PT, and an overexpression of *LINGO2*, *SEC22C*, and *VAMP3* at 48 h PT. A significant positive correlation of expression with *LINC00518* was observed for *SEC22C* and *VAMP3* (Figure 8, Appendix A). In MEL270 cells, IL6 treatment increased the expression of *LINGO2*, *OTUD7B,* and *VAMP3* at both 24 and 48h, *NFIA* at 48 h and *SEC22C* at 24 h PT (Figure 8, Appendix A). *LINGO2* and *OTUD7B* also showed a significant positive expression correlation with *LINC00518* after IL6 treatment (Figure 8, Appendix A).

The same analysis was also carried out in UM cells treated with CoCl_2_ to induce a hypoxia-like response mediated by *HIF1A* activation. In 92.1 cells, all mRNA targets, excluding *VAMP3*, were upregulated: in particular, *LINGO2* and *OTUD7B* showed increased levels at both 24 and 48 h PT, while *NFIA*, and *SEC22C* levels raised at 48 h PT. All targets showed a significant and strong positive expression correlation with *LINC00518* (Figure 9, Appendix A). CoCl_2_ treatment in MEL270 confirmed these results, inducing increased expression of *LINGO2* and *NFIA* at 24 and 48h, and *OTUD7B*, *SEC22C*, and *VAMP3* at 48 h PT (Figure 9, Appendix A). Significant positive expression correlation with *LINC00518* was confirmed for *SEC22C* and *VAMP3* (Figure 9, Appendix A).

### 2.6. RNA–RNA Interaction between LINC00518 and mRNAs

Based on the miRNA sponge model, the reduced expression of *LINGO2*, *NFIA*, *OTUD7B*, *SEC22C*, and *VAMP3* may be induced by *LINC00518*-mediated sponging of miRNAs. Another hypothesis explaining this regulatory mechanism is the “miRNA protector” model [17]. According to this hypothesis, lncRNAs may directly bind the 3′-UTR of the mRNAs, thus preventing miRNA-mRNA interactions and impairing miRNA-mediated silencing. To investigate this hypothesis, we computationally analyzed the RNA–RNA interactions between *LINC00518* and the 3′-UTR of the five mRNAs. This analysis showed several regions within the 3′-UTRs characterized by strong negative free energy (NFE), suggesting a very effective interaction with *LINC00518*. Some of these regions overlapped with binding sites of the potentially sponged miRNAs analyzed in this study (Figure 10). This result suggests that *LINC00518* may inhibit miRNA regulative function by binding and masking miRNA-binding sites on the 3′-UTR of mRNAs.

### 2.7. Identification of Transcription Factors Potentially Regulating LINC00518 Expression

To investigate the transcriptional regulation underlying *LINC00518* overexpression in UM tumor tissue, we queried the UCSC Genome Browser to retrieve the transcription factors (TFs) with experimentally validated transcription factor binding sites (TFBSs) in correspondence with or upstream of the transcription start site of the lncRNA locus. For each identified TF, correlation of expression with *LINC00518* was investigated in the UM TCGA dataset. Aiming to identify the potential activator of *LINC00518* expression, we considered TFs showing a significant positive correlation of expression with the lncRNA in the UM-TCGA dataset, namely, *KDM1A* (lysine demethylase 1A) and *MITF* (melanocyte inducing transcription factor), a TF specifically expressed in melanocytes (Figure 11 and Appendix A).

To investigate the transcriptional control performed by *KDM1A* on *LINC00518*, we blocked TF activity by treating 92.1 cells with the inhibitor ORY-1001. Expression of *LINC00518* was evaluated, together with *CDH1* and *VEGFA*, two known targets of *KDM1A* in other cancer models. The inhibitor efficiently induced an increased expression of *CDH1*, physiologically repressed by *KDM1A*, and a reduction in *VEGFA* levels, whose expression is activated by the TF, confirming the efficiency of the treatment, particularly after 48 h. *LINC00518* expression showed a slight decrease at 48h, which did not reach significance (Figure 11).

*MITF* inhibition was achieved through A-485 treatment. The efficiency of *MITF* inhibition was evaluated by analyzing mRNA levels of *MITF* itself and *TYR* (tyrosinase), transcriptional target of *MITF*. In 92.1 cells, both *MITF* and *TYR* showed a strong reduction of expression at both 24 and 48 h PT, proving that A-485 effectively inhibited *MITF* expression and, consequently, its activity. Reduced expression of *LINC00518* was observed at 24 h PT. Moreover, *LINC00518* showed a significant positive expression correlation with *MITF* (Figure 11). Analysis on MEL270 cells confirmed these data, with a smaller reduction in *LINC00518* levels that persisted at 48 h PT (Figure 11). These results support the hypothesis that *MITF* acts as a transcriptional activator of *LINC00518* expression. *MITF* expression was also evaluated in UM biopsies, confirming its increased expression in tumor tissue (fold change = 25.5, *p*-value = 0.002); the significant positive expression correlation between *LINC00518* and *MITF* mRNA was confirmed (r-value = 0.78, *p*-value ≤ 0.001), strengthening our hypothesis also in patient tissues.

## 3. Discussion

The crucial role played by lncRNAs in carcinogenesis is today widely accepted. Nevertheless, understanding the molecular mechanisms regulated by lncRNAs is very difficult, since they demonstrated a high heterogeneity of functions. One of the most accredited hypotheses is that lncRNAs participate in complex networks based on competitive interactions among different RNA molecules, including lncRNAs, miRNAs, and mRNAs [7]. The aim of our study was to investigate the RNA-based network underlying biological processes associated with UM. First, we selected a set of lncRNAs from the expression data available in TCGA. Unfortunately, this dataset only includes tumor tissue samples; therefore, we were not able to identify lncRNAs showing differential expression in tumors compared to normal tissues. To overcome this obstacle, we exploited *MET* and *BAP1*, acting, respectively, as oncogene and tumor suppressor genes in UM, as “decoy” genes and identified lncRNAs showing an opposite trend of expression correlation with each of the two genes. By this approach, we identified 11 lncRNAs and analyzed their expression in tumors compared to normal adjacent tissues obtained from enucleated eyes of 41 UM patients. This analysis showed the upregulation of two lncRNAs, namely, *LINC00518* and *LINC00634*, suggesting their potential oncogenic participation in UM. Some reports have already suggested an oncogenic role for *LINC00518* in several other cancer models, including breast cancer [18,19], cervical cancer [20], prostate cancer [21], and skin melanoma [22]. *LINC00518* was associated with skin melanoma and its expression levels were used to discriminate melanomas from nevi in a non-invasive patch test developed to evaluate pigmented skin lesions [23,24,25,26,27]. On the contrary, little is known about *LINC00634*, with only one recent paper reporting its oncogenic role in esophageal squamous cell carcinoma [28]. Although *LINC00518* had no statistical association with clinicopathological parameters of patients, it showed the strongest deregulation. Therefore, we focused our attention on its potential involvement in UM biology. *LINC00518* exhibited a preferential cytoplasmic localization but was undetectable in serum and vitreous humor, as well as in exosomes isolated from these fluids. Based on these results, we hypothesized that *LINC00518* may play a crucial role in tumor progression, leading tumor cells to maintain this lncRNA within their cytoplasm, where it performs its function. Artificial activation/block of cell cycle achieved by serum supplement/starvation and MAPK inhibition showed no alteration of *LINC00518* expression, suggesting that the expression of this lncRNAs is not induced by cell cycle progression in UM.

On the contrary, functional assays aimed to investigate its implication in metastatic processes showed that *LINC00518* expression increased not only after EMT-induction by IL6 and HGF treatment, but also after *HIF1A* activation by CoCl_2_ treatment. It has been reported that IL6 and HGF trigger EMT in vitro [29,30,31,32,33,34], also in UM cells [35,36], while CoCl_2_ provokes a hypoxia-like response through *HIF1A* activation [37], which represents one of the key events in the metastatic cascade [38]. Based on our results, IL6 seems to be more efficient in EMT induction in UM cells compared to HGF, as confirmed by the higher increase in EMT markers. In agreement with this, *LINC00518* showed a lower increase of expression in HGF-treated 92.1 cells compared to the same cells treated with IL6. However, HGF failed to induce EMT in MEL270 even if administered at higher concentrations than in 92.1; hypothetically, this difference in HGF efficiency may be due to the different genetic and epigenetic background specific for each cell line. Overexpression of *LINC00518* was also induced by the *HIF1A*-induced hypoxia-like response. Taken together, these data suggest that the oncogenic role of *LINC00518* may be performed in the initiation of the metastatic process. This hypothesis was confirmed by transient silencing of *LINC00518*, which showed a decreased migration ability of UM cell lines when *LINC00518* was down-modulated. Metastasization is a severe aggravation of the disease, affecting nearly half of UM patients, for which no therapy is available [39]; therefore, *LINC00518* could represent an important target for new RNA-based antitumor therapeutic approaches. The involvement of *LINC00518* in metastasization was also reported in other cancer models [18,20], including skin melanoma [22]. Viability assays showed that reduced *LINC00518* expression also decreased cell proliferation in both UM cell lines. This result suggests that *LINC00518* expression, albeit not controlled by cell cycle modulation, could indirectly induce cell cycle progression, thus fostering tumor growth. Based on these considerations, we suggest a potential double oncogenic function for *LINC00518* in UM, performed by promoting cell proliferation as well as migration.

In light of its preferentially cytoplasmic localization, supporting the ceRNA network hypothesis, we investigated the RNA-based network related to *LINC00518*. We hypothesized that *LINC00518* may act as a miRNA sponge, binding miRNAs and thus preventing them from negatively regulating their target mRNAs [17]. Several papers reported the miRNA sponge function for this lncRNA in various cancer models [19,21], including skin melanoma [22], supporting our hypothesis. The expression of tumor suppressive miRNAs potentially sponged by *LINC00518* showed no significant alteration in both cell lines after lncRNA silencing. This result is not surprising since the miRNA sponge function of lncRNAs determines a reduced availability of free miRNAs in the cytoplasm [40], while no influence on miRNA stability has been described. More importantly, some of the mRNAs targeted by the potentially sponged miRNAs showed a significantly reduced expression and/or a significant positive expression correlation with *LINC00518* after its transient silencing. In particular, five mRNAs showed significant results in both cell lines: *LINGO2*, *NFIA*, *OTUD7B*, *SEC22C,* and *VAMP3*. These findings would suggest that LINC00518 may indirectly exert its oncogenic function by protecting *LINGO2*, *NFIA*, *OTUD7B*, *SEC22C,* and *VAMP3* from miRNA binding and degradation. Oncogenic functions have been reported for most of these five genes and their expression increased in our cohort of UMs compared to normal tissues. *LINGO2* (leucine rich repeat and Ig domain containing two) encodes a transmembrane protein that was recently reported as a cancer stem cell (CSC) marker and oncogene in gastric cancer; high expression of *LINGO2* was associated with increased cell motility, angiogenic capacity, and tumorigenicity [41]. *NFIA* (nuclear factor I A) encodes a member of the NF1 (nuclear factor 1) family of TFs known to act as oncogenes in several cancer models [42,43,44]. *OTUD7B* (OTU deubiquitinase 7B), also known as *CEZANNE*, encodes a multifunctional deubiquitinylase enzyme that is essential in inflammation and proliferation signals. Its increased expression has been reported in lung squamous carcinoma and adenocarcinoma compared to normal tissue; in the same cancer model, *OTUD7B* promoted cell proliferation, migration and metastasization [45]. Moreover, a lncRNA-miRNA-mRNA axis promoting tumor progression and involving *OTUD7B* has been reported in pancreatic cancer [46]. *SEC22C* (SEC22 homolog C, vesicle trafficking protein) encodes a member of the SEC22 family of vesicle trafficking proteins. No specific association with cancer-related processes has been reported to date for this gene; however, it has been demonstrated that vesicle trafficking plays a crucial role in tumorigenesis [47]. Moreover, its homolog *SEC22B* (SEC22 homolog B, vesicle trafficking protein) is involved in carcinogenesis and harbors several mutations associated with different cancer models [48]. *VAMP3* (vesicle associated membrane protein 3) encodes a member of the vesicle-associated membrane protein (VAMP)/synaptobrevin family and is involved in integrin trafficking and cell migration. Indeed, silencing of *VAMP3* impaired cell migration and integrin-mediated adhesion in pancreatic cancer cells [49]. *VAMP3* knockdown also decreased invasiveness in vitro by affecting vesicle cargo and delivery in melanoma cells [50].

In addition to the miRNA sponge function, we also proposed a “miRNA protector” model involving *LINC00518* and the target mRNAs [17]. The transcript levels of *LINGO2*, *NFIA*, *OTUD7B*, *SEC22C*, and *VAMP3* may be controlled not only by *LINC00518*-mediated miRNA sponging, but also by a direct interaction between the lncRNA and the 3′-UTR of the mRNAs, resulting in the masking of miRNA-binding sites. It is important to note that these interactions would affect the binding of the miRNAs here analyzed as potentially sponged by *LINC00518*, as well as many other miRNAs that we have not considered in this analysis. Therefore, both the models would explain the positive expression correlation between lncRNA and mRNAs, preventing miRNA-mediated degradation.

We also investigated the regulative mechanism underlying increased *LINC00518* expression in UM. By using experimental data from the ENCODE Project and correlation data computed on the UM TCGA dataset, we identified *KDM1A* and *MITF* as hypothetical transcriptional activators of *LINC00518* and performed in vitro inhibition of their activity. Our results suggest that increased *LINC00518* expression may be promoted by *MITF*, a TF specific for melanocytes and essential for their development, also being responsible for melanogenesis regulation [51] and associated with UM. We observed an increased *MITF* expression and a positive expression correlation with *LINC00518* in our cohort of UM patients. Previous studies reported *MITF* overexpression in UM cell lines compared to normal melanocytes and a cell cycle block at the G1 phase induced by *MITF* silencing [52]. This evidence, together with the decreased lncRNA expression induced by the inhibition of TF activity, supports the potential role of *MITF* in the induction of *LINC00518* expression in UM tissue. The potential involvement of *MITF* in *LINC00518* regulation suggests a specific expression of this lncRNA in melanocytes. Indeed, an RNA-seq study on 27 human tissues showed high levels of *LINC00518* in skin, placenta, and testis, while other tissues exhibited low expression of the lncRNA [53].

## 4. Materials and Methods

### 4.1. LncRNA Selection

Selection of lncRNAs potentially involved in UM molecular processes was performed by using transcriptomic data available at TCGA [54]. The TCGA UM dataset consists of expression data obtained from 80 samples of tumor tissues but does not include non-tumoral samples. To obtain a list of potential UM-associated lncRNAs, we retrieved from R2: Genomics Analysis and Visualization Platform (http://r2.amc.nl, accessed on 9 November 2020) the list of transcripts showing expression correlation with the mRNA of two genes known to be involved in UM, used as “decoys”: *MET*, with an oncogenic function [55], and *BAP1*, with a tumor suppressive function. Among the transcripts significantly correlated with the decoys, we selected the long intergenic RNAs (those transcripts named “LINC-”) and the antisense transcripts (named “-AS1”). Comparing the two lists of *MET* and *BAP1*-correlated lncRNAs, we selected those transcripts showing strong correlations and *p*-values and, preferably, an opposite correlation trend with each of the two decoys. Following this approach, we identified potential oncogenes (positive correlation with *MET* and negative with *BAP1*) and potential tumor suppressor genes (positive correlation with *BAP1* and negative with *MET*).

### 4.2. Patient Recruitment and Tissue, Vitreous, and Serum Sampling

Forty-one patients affected by uveal melanoma were enrolled at the Eye Clinic of the University of Catania; according to the Declaration of Helsinki, all participants signed a written informed consent. All experiments were approved by the local Ethics Committee, Comitato Etico Catania 1, University of Catania (ID: 003186-24). Clinicopathological data of study participants are shown in Appendix A. From each patient, FFPE tumor and normal tissue samples, vitreous humor, and serum were collected. Enucleations were performed at the Eye Clinic of the University of Catania; after surgery, enucleated eyes were formalin-fixed and paraffin-embedded and successively analyzed for tumor staging at the Anatomical Pathology Unit, Department G.F. Ingrassia, University of Catania. Vitreous humor and serum were obtained from UM patients, and circulating exosomes were isolated from both fluids as previously reported [8]. Exosomes were isolated from vitreous humor and serum through ExoQuick (System Biosciences, Palo Alto, CA, USA); according to the manufacturer’s instructions, 400 µL serum/vitreous were mixed with 100 µL ExoQuick. The exosome pellet was directly lysed with Qiazol Lysis Reagent (Qiagen, Hilden, Germany) for total RNA isolation.

### 4.3. RNA Isolation from FFPE Tissues, Vitreous, and Serum Samples

Total RNA was isolated from 10 µm slices (0.01 g of weight and 1.5–2.5 cm of diameter each) of tumor tissue and natural adjacent tissue by using the PureLink FFPE RNA Isolation Kit (Invitrogen, Thermo Fisher Scientific, Waltham, MA, USA), according to the manufacturer’s instructions. Total circulating RNA was isolated from vitreous and serum (400 µL) by the miRNeasy Mini-Kit (Qiagen), according to the Qiagen supplementary protocol for total RNA isolation from serum and plasma. MiRNeasy Mini Kit was also used for total RNA isolation from vitreous and serum exosomes. RNA was quantified through a Qubit fluorimeter (Invitrogen).

### 4.4. Subcellular Fractionation

The RNA Subcellular Isolation kit (Active Motif, Carlsbad, CA, USA) was used to analyze *LINC00518* expression in the nucleus and cytoplasm of 92.1 and MEL270 cells. The lncRNA *MALAT1* and the circular RNA *CDR1-AS* were used as markers of nuclear and cytoplasmic fractions, respectively. Expression analysis was performed by Real-Time PCR.

### 4.5. Computational Analysis

To investigate the involvement of *LINC00518* in UM molecular processes, we identified its potential molecular axes, composed of miRNAs and mRNAs, through a computational analysis performed by using miRNA and mRNA expression data deposited in the UM dataset of TCGA [54]. The approach we used is based on the “miRNA sponge” hypothesis, according to which a lncRNA may bind and sequester miRNAs, impairing negative regulation of target mRNA translation or stability [7]. According to this model, if the lncRNA acts as an oncogene, thus showing increased expression in tumor tissue compared to control tissue, its “sponge” function would protect mRNAs acting as oncogenes too, hence showing positive correlation of expression with the lncRNA. Accordingly, lncRNA would sequester miRNAs showing negative correlation of expression with both lncRNA and mRNAs. In agreement with this hypothesis, miRNAs showing a significant negative correlation of expression with *LINC00518* were identified by computing the Pearson correlation coefficient; similarly, mRNAs with significant positive correlation of expression with *LINC00518* were determined. Successively, the top 50 negatively correlated miRNAs were investigated for the presence of miRNA-binding sites within *LINC00518* sequence by using RNA22 (https://cm.jefferson.edu/rna22/, accessed on 9 November 2020): miRNAs showing multiple binding sites, low *p*-values and free energy values were selected as potentially sponged by *LINC00518*. To identify the mRNAs targeted by these potentially sponged miRNAs, two parallel approaches were followed. The first one led to the identification of potential miRNA targets based on anticorrelation of expression: using the UM TCGA dataset, all mRNAs showing significant negative correlation with each miRNA were identified. This list was then compared with that of mRNAs showing significant positive correlation of expression with *LINC00518*. The mRNAs included in both lists, characterized by a positive correlation with *LINC00518* and a negative correlation with the miRNA, were selected. On the other hand, the second approach for miRNA target identification was based on sequence complementarity: the miRWalk 3.0 (mirwalk.umm.uni-heidelberg.de/, accessed on 09/11/2020) tool was used to retrieve all the mRNAs showing multiple binding sites within the 3′-UTR region for each miRNA. The lists obtained following these two approaches were compared and the shared transcripts were selected, obtaining a list of mRNAs characterized by (i) a positive correlation of expression with *LINC00518*, and (ii) a negative correlation of expression with the miRNAs potentially sponged by the lncRNAs and harboring multiple binding sites on *LINC00518* sequence. This final list was further filtered by using RNA22 data, thus selecting those mRNAs having also the highest probability of being bound by miRNAs.

To investigate the transcriptional mechanisms underlying abnormal expression of *LINC00518*, we used the UCSC Genome browser (https://genome.ucsc.edu/, accessed on 9 November 2020) to retrieve the TFBSs experimentally identified within the promoter and the upstream regulatory region (1 kb) of the *LINC00518* gene locus. More specifically, we used the ENCODE tracks (i.e., DNase hypersensitive and chromatin immunoprecipitation sequencing (ChIP-seq) on transcription factors, and H3K27Ac, H3K4me1, and H3K4me3 histone marks) retrieving a list of TFs potentially regulating *LINC00518* expression. These TFs were then investigated for correlation of expression with *LINC00518* in the UM TCGA dataset.

RNA-RNA interactions between *LINC00518* and mRNAs were predicted by the IntaRNA tool (https://rna.informatik.uni-freiburg.de/IntaRNA/Input.jsp, accessed on 9 November 2020) [56]. Positions of miRNA-binding sites on mRNA 3′-UTR were retrieved from the miRWalk outputs.

### 4.6. In Vitro Functional Assays on UM Cells

To investigate the involvement of *LINC00518* in UM molecular processes, we performed functional assays on two UM cell lines: 92.1 and MEL270. In particular, we investigated the role of *LINC00518* in (i) cell cycle progression, by performing both MAPK pathway inhibition and cell cycle block/activation induced by serum starvation/supplement; (ii) metastatic processes, by inducing EMT; (iii) CoCl_2_-induced hypoxia-like response, by inducing *HIF1A* activation. Finally, siRNA-mediated transient silencing of *LINC00518* was performed to evaluate the effects of its reduced expression on its molecular axes and cell phenotype. We also investigated the role of TFs *KDM1A* and *MITF* in the regulation of *LINC00518* expression by chemically inhibiting their function in UM cell lines.

Cell cultures

Human UM cell line 92.1 was cultured in RPMI-1640 medium (Gibco, Thermo Fisher Scientific, Waltham, MA, USA) supplemented with 10% fetal bovine serum (FBS) (Gibco) and 2 mM L-glutamine (Lonza, Basel, Switzerland). Medium was supplemented with 1% penicillin/streptomycin (10,000 U/mL) (Gibco). The same medium supplemented with 1% sodium pyruvate and 1% non-essential amino acids was used for MEL270. Cells were cultured at 37 °C and 5% CO_2_.

MAPK inhibition

MAPK pathway inhibition was performed by treating UM cells with U0126 (MEK1/2 inhibitor, Merck, Kenilworth, NJ, USA), a highly selective ATP-non-competitive inhibitor of both Mek1 and Mek2 kinases. Cells (3 × 10^4^ per well) were seeded in 24-well plates and grown in serum starvation (no FBS) for 24 h prior to treatment with 12.5 µM or 25 µM inhibitor; cells used as controls were treated with an equal volume of DMSO, the solvent used for the inhibitor solubilization. Cell viability was assessed by the MTT assay at 3, 6, 12, and 24 h PT (8 × 10^3^ cells/well), aiming to identify the most efficient concentration of the inhibitor. Gene expression was analyzed at 3, 6, 12, 24, and 48 h after treatment. All experiments were performed in biological triplicates.

Cell cycle block and activation

Cell cycle arrest and activation were induced by removing and adding serum to cell media, respectively. Cells (5 × 10^4^ per well) were seeded in 24-well plates. To activate cell cycle, cells were first grown in serum starvation (no FBS) for 24 h; after that, serum starvation was maintained in control cells, while treated cells were supplemented with 10% FBS, thus activating cell cycle. *LINC00518* expression was evaluated after 30 min, 1 and 2 h from serum supplement, in order to investigate its involvement in early responses to serum-induced cell cycle activation. To confirm treatment efficiency, expression of *SRF* mRNA was also evaluated at the same time points. Additionally, we investigated *LINC00518* expression after 24 h from serum supplement or deprivation, assessing its role in delayed response to serum-induced cell cycle activation and block, respectively. For cell cycle activation, we used the same protocol mentioned for early stages; to block cell cycle, treated cells were deprived of serum for 24 h, inducing transition to the G0 phase and cell cycle arrest, while control cells were maintained in 10% FBS medium.

EMT induction

EMT was induced in two different experiments, by growing cells in the presence of IL6 (Gibco) (200 ng/mL for 92.1 and 300 ng/mL for MEL270) and HGF (Sigma-Aldrich, St. Louis, MO, USA) (50 ng/mL for 92.1 and 100 ng/mL for MEL270). Cells (3 × 10^4^ per well) were seeded in 24-well plates and grown in serum starvation (no FBS) for 24 h prior to treatment. Cells used as controls were treated with an equal volume of acetic acid, the solvent used for IL6 solubilization, or water, used to dissolve HGF. Gene expression analysis was performed at 24 and 48 h PT. All experiments were performed in biological triplicates. To evaluate treatment efficiency, mRNA expression of EMT markers *CDH1*, *VIM*, and *ZEB2* was evaluated.

*HIF1A* activation

*HIF1A* activation was induced by treating cells with CoCl_2_ (Sigma-Aldrich). Cells (3 × 10^4^ per well) were seeded in 24-well plates and treated with 100 µM (92.1) or 150 µM (MEL270) CoCl_2_. Cells used as controls were treated with an equal volume of PBS (phosphate-buffered saline). Gene expression analysis was performed at 24 and 48 h PT. All experiments were performed in biological triplicates. To evaluate treatment efficiency, mRNA expression of *VEGFA*, target of *HIF1A*, was evaluated.

*LINC00518* transient silencing

Silencing of *LINC00518* was achieved by using Silencer Select siRNAs (Thermo Fisher Scientific, Waltham, MA, USA), according to the manufacturer’s instructions. Cells (5 × 10^4^ per well) were seeded in 24-well plates and reverse-transfected with 10 nM (92.1) or 20 nM (MEL270) siRNA using Lipofectamine RNAiMAX (Thermo Fisher Scientific) as transfectant agent. All experiments were performed in biological triplicates. The expression of *LINC00518* and its targets was evaluated at 24 and 48 h post-transfection. Replicates showing a transfection efficiency lower than 75% were excluded from analysis. The effects of *LINC00518* silencing on cell proliferation and migration were evaluated through the MTT assay and Oris Cell Migration Assay (Platypus Technologies, Fitchburg, WI, USA), respectively. For the MTT assay, cells (1 × 10^4^ per well) were seeded, and reverse transfected in 96-well plates, and cell viability was evaluated 24 and 48 h post-transfection. For the migration assay, cells (1 × 10^4^ per well) were seeded, and reverse transfected in an Oris Cell Migration Assay 96-well plate after insertion of Oris Cell Seeding Stoppers. Stoppers were removed 24 h after transfection (T0) and migration was evaluated at different time points (24, 48, 72, 96, and 144 h after stopper removal). Cells migrated in the detection zone were quantified by using ImageJ software. All experiments were performed in six biological replicates for both proliferation and migration assays. To investigate *LINC00518* network, the expression of miRNAs and mRNAs selected as potential interactors of the lncRNA was analyzed. Selection of mRNAs considered as potential targets of *LINC00518* was carried out according to two evidence: (i) decreased expression of the mRNA after *LINC00518* transient silencing, suggesting a regulation operated by the lncRNA; (ii) the conservation of positive correlation of expression with *LINC00518* in cells where the lncRNA expression is transiently reduced, which suggests an involvement of *LINC00518* in the regulation of mRNA expression. Moreover, mRNAs were considered as potential *LINC00518* targets only if statistically significant results in expression modulation or correlation were observed in both cell lines.

Inhibition of TFs potentially regulating *LINC00518* expression

TF inhibition was achieved by chemical inhibitors rather than siRNA transfection in order to avoid potential off-target effects. *KDM1A* activity was inhibited by treating UM cell lines with ORY-1001 (TargetMol, Boston, MA, USA), while inhibition of *MITF* was achieved by A-485 (MedChemExpress, Princeton, NJ, USA), which induces a reduction in both mRNA and protein levels of *MITF* [57]. Cells (3 × 10^4^ per well) were seeded in 24-well plates and exposed to 120 nM ORY-1001 and 5 µM (92.1) or 10 µM (MEL270) A-485 for 24 and 48 h. Cells used as controls were treated with an equal volume of DMSO, the solvent used for inhibitor solubilization. All experiments were performed in biological triplicates. To assess the efficiency of *KDM1A* inhibition, we evaluated mRNA expression of its targets *CDH1* and *VEGFA*, respectively repressed and activated by the TF in other cancer models [58,59]. For *MITF* inhibition, mRNA levels or the TF itself and its target *TYR* were evaluated.

### 4.7. Expression Analysis through Real-Time PCR

To evaluate the expression of lncRNAs and mRNAs, PCR primers specific for the intended transcripts were designed using PrimerBlast (https://www.ncbi.nlm.nih.gov/tools/primer-blast/, accessed on 9 November 2020) (Appendix A). Expression of lncRNAs and mRNAs was investigated by using Power SYBR Green RNA-to-CT 1-Step kit (Applied Biosystems, Thermo Fisher Scientific, Waltham, MA, USA) according to the manufacturer’s instructions. MiRNA expression was evaluated by using TaqMan microRNA assays: miRNA-specific cDNA was retrotranscribed from total RNA through the TaqMan microRNA Reverse Transcription Kit (Applied Biosystems) and then amplified with TaqMan Universal Master Mix II, no UNG (Applied Biosystems). All Real-Time PCR reactions were performed on a 7900HT Fast Real-Time PCR System (Applied Biosystems). Differentially expressed (DE) transcripts were identified through the SDS RQ Manager 1.2 software (Applied Biosystems); normalization was performed using *ACTB* (actin beta) for biopsies, *HPRT1* (hypoxanthine phosphoribosyltransferase 1) for cell lines, and *RNU6* (RNA, U6 small nuclear 1) for miRNA analysis as endogenous controls. Expression fold changes of DE lncRNAs, mRNAs and miRNAs were calculated by applying the 2^−ΔΔCt^ method.

### 4.8. Statistical Analysis

Statistical analysis of Real-Time PCR expression data was performed by evaluating normality and homogeneity of variance of ΔCt distributions with GraphPad Prism 8 software; according to the results, a parametric (homoscedastic or Welch corrected) or non-parametric *t*-test was applied. The existence of correlation between lncRNA expression (expressed as fold change) and clinicopathological parameters was investigated. For numerical variables, correlation was evaluated by calculating the Pearson or Spearman correlation coefficient, according to normality of distributions of fold changes and clinicopathological parameters; the same approach was followed to assess correlation of expression between different transcripts, using ΔCts as input data. For categorical variables, samples were divided into groups according to the considered feature and differential expression between these groups was evaluated by applying *t*-test or ANOVA test, parametric or non-parametric, according to normality of the obtained distributions.

## 5. Conclusions

We identified *LINC00518* as a new oncogenic lncRNA in UM. Increased *LINC00518* expression may be induced by *MITF*, a melanocyte-specific TF, and promote cancer-related processes, such as cell proliferation and migration. According to our hypothesis, *LINC00518* regulates a ceRNA network including both miRNAs and mRNAs. Specifically, after *LINC00518* transient silencing, we observed reduced levels of *LINGO2*, *NFIA*, *OTUD7B*, *SEC22C*, and *VAMP3* mRNAs, previously reported as oncogenes in several cancer models and targeted by tumor suppressive miRNAs potentially sponged by *LINC00518*. We also proposed that the direct binding of LINC00518 to the 3′-UTR of these mRNAs may prevent miRNA-mediated negative regulation. Overall, this study suggests a potential miRNA sponge/miRNA protector model for LINC00518 by showing its systemic effects on multiple miRNAs and mRNAs, without focusing on specific LINC00518:miRNA interactions, which we plan to analyze in further studies. Our results suggest that *LINC00518* may be investigated as a potential target for new RNA-based therapeutic approaches against UM.

## Figures and Tables

**Figure 1 cancers-12-03867-f001:**
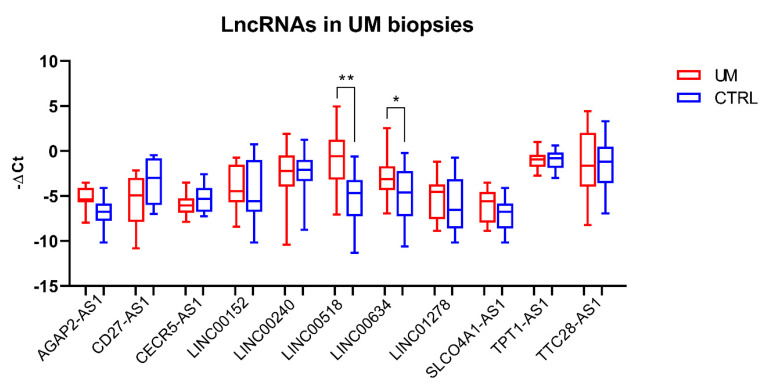
LncRNA expression analysis in tissue biopsies. Analysis was performed through Real-Time PCR in 41 pairs of tumor and normal tissues. *: *p*-value < 0.05; **: *p*-value < 0.0001.

**Figure 2 cancers-12-03867-f002:**
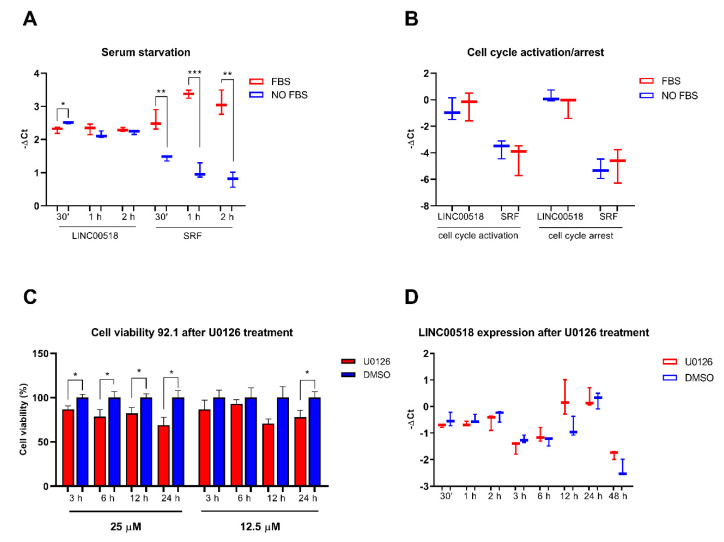
Functional assays aiming to evaluate *LINC00518* expression induced by cell cycle in 92.1 cells. (**A**) Expression of *LINC00518* during early response to serum-induced cell cycle activation. (**B**) Expression of *LINC00518* at 24 h after serum supplement (cell cycle activation) and serum deprivation (cell cycle arrest). (**C**) Evaluation of cell viability after U0126 treatment of 92.1 cells through MTT assay; the effects of two concentrations were tested, showing that 25 µM induced the most significant decrease in cell viability in U0126-treated cells compared to control cells (DMSO). (**D**) Expression of *LINC00518* in U0126-treated cells compared to control cells (DMSO). Each experiment was performed in three in vitro replicates. *: *p*-value < 0.05; **: *p*-value < 0.005; ***: *p*-value < 0.0005.

**Figure 3 cancers-12-03867-f003:**
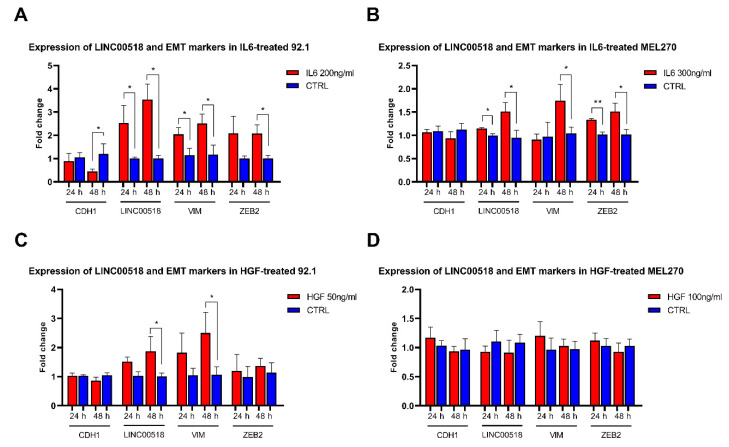
Expression of epithelial to mesenchymal transition (EMT) markers and *LINC00518* in uveal melanoma (UM) cells treated with interleukin 6 (IL6) and hepatocyte growth factor (HGF). (**A**) 92.1 cells treated with IL6. (**B**) MEL270 cells treated with IL6. (**C**) 92.1 cells treated with HGF. (**D**) MEL270 cells treated with HGF. Each experiment was performed in three in vitro replicates. *: *p*-value < 0.05; **: *p*-value < 0.005.

**Figure 4 cancers-12-03867-f004:**
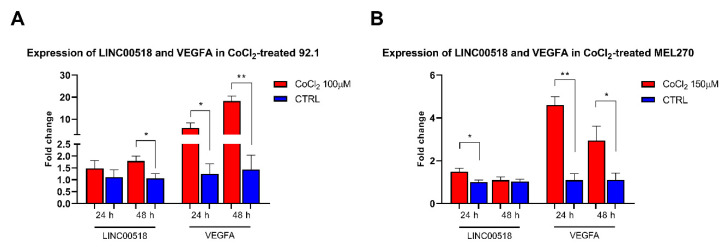
Expression of *VEGFA* and *LINC00518* after *HIF1A* activation. (**A**) 92.1 cells treated with CoCl_2_. (**B**) MEL270 cells treated with CoCl_2_. *: *p*-value < 0.05; **: *p*-value < 0.0005.

**Figure 5 cancers-12-03867-f005:**
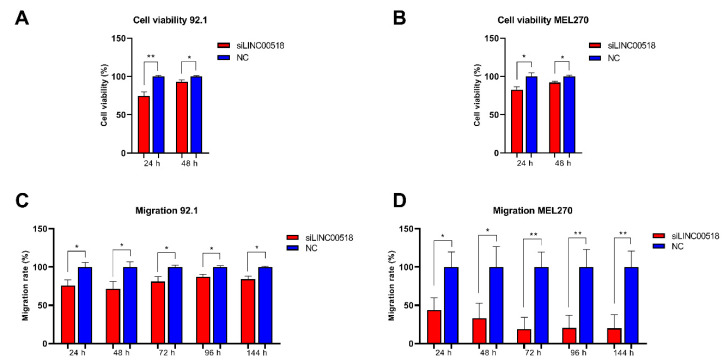
Evaluation of cell viability and migration after *LINC00518* transient silencing. Cell viability was assessed through the MTT assay in (**A**) 92.1 and (**B**) MEL270. Migration was investigated by the Oris Cell Migration Assay in (**C**) 92.1 and (**D**) MEL270. Each experiment was performed in six in vitro replicates. *: *p*-value < 0.05; **: *p*-value < 0.005.

**Figure 6 cancers-12-03867-f006:**
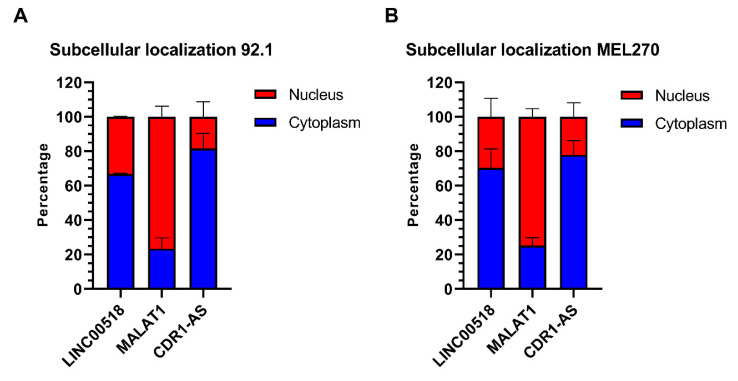
Subcellular localization of *LINC00518*. Expression analysis in (**A**) 92.1 and (**B**) MEL270 cells. *MALAT1* and *CDR1-AS* were used as markers of nuclear and cytoplasmic fractions, respectively. Data are shown as percentages. Analysis was performed in three in vitro replicates.

**Figure 7 cancers-12-03867-f007:**
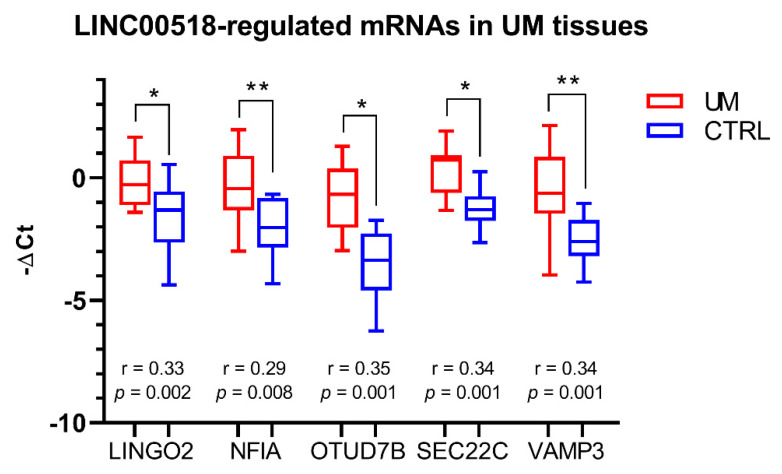
Expression of *LINC00518*-regulated mRNAs in UM biopsies. For each mRNA, the r-value and the associated *p*-value measuring correlation of expression with *LINC00518* are reported. Analysis was performed in 41 pairs of tumor and normal tissues. *: *p*-value < 0.05; **: *p*-value < 0.005.

**Figure 8 cancers-12-03867-f008:**
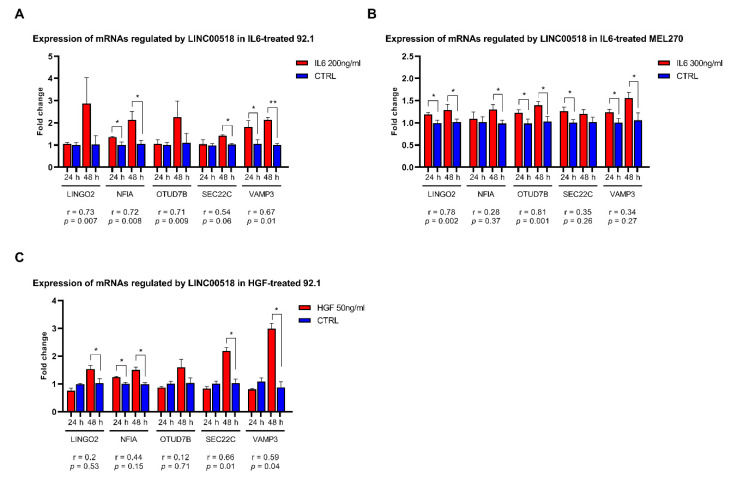
Expression of mRNAs regulated by *LINC00518* after EMT induction in UM cells. Expression analysis in: (**A**) IL6-treated 92.1 cells; (**B**) IL6-treated MEL270 cells; (**C**) HGF-treated 92.1 cells. For each mRNA, the r-value and the associated *p*-value measuring correlation of expression with *LINC00518* are reported. Each experiment was performed in three in vitro replicates. *: *p*-value < 0.05; **: *p*-value < 0.005.

**Figure 9 cancers-12-03867-f009:**
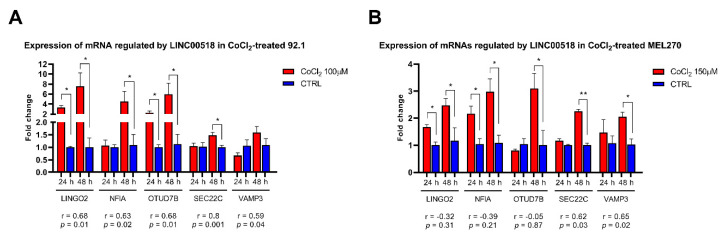
Expression of mRNAs regulated by *LINC00518* after *HIF1A* activation in UM cells. For each mRNA, the r-value and the associated *p*-value measuring correlation of expression with *LINC00518* are reported. Each experiment was performed in three in vitro replicates. *: *p*-value < 0.05; **: *p*-value < 0.005.

**Figure 10 cancers-12-03867-f010:**
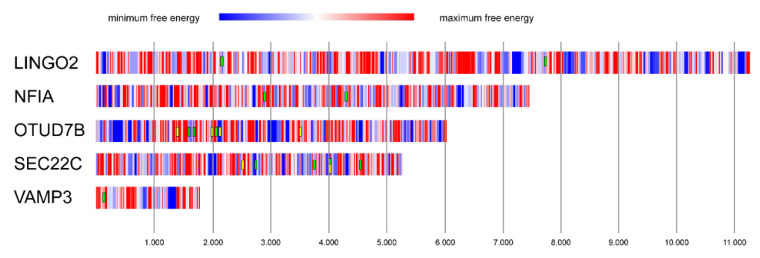
Prediction of RNA–RNA interaction between *LINC00518* and mRNAs. Graphical representation of free energy (kJ) computed from the binding of *LINC00518* and the 3′-UTRs of the regulated mRNAs. A color-coded scale, based on free energy values, is used to depict the 3′-UTRs of mRNAs: blue regions represent the binding tracts with minimum free energy, and, accordingly, they represent the most reliable binding regions. The boxes represent miRNAs binding to minimum free energy regions: *LINGO2*: miR-497-5p; *NFIA*: miR-143-3p; *OTUD7B*: miR-145-5p (green) and miR-497-5p (yellow); *SEC22C*: miR-199a-5p (green) and miR-212-5p (yellow); *VAMP3*: miR-3191-3p. Free energy scale limits: (1) *LINGO2*: minimum = −28.57, maximum = 2.32; (2) *NFIA*: minimum = −20.18, maximum = 2.14; (3) *OTUD7B*: minimum = −33.26, maximum = 8.07; (4) *SEC22C*: minimum = −27.55, maximum = 1.09; and (5) *VAMP3*: minimum = −23.87, maximum = −0.59. Nucleotide position of the 3′UTR of each mRNA is shown.

**Figure 11 cancers-12-03867-f011:**
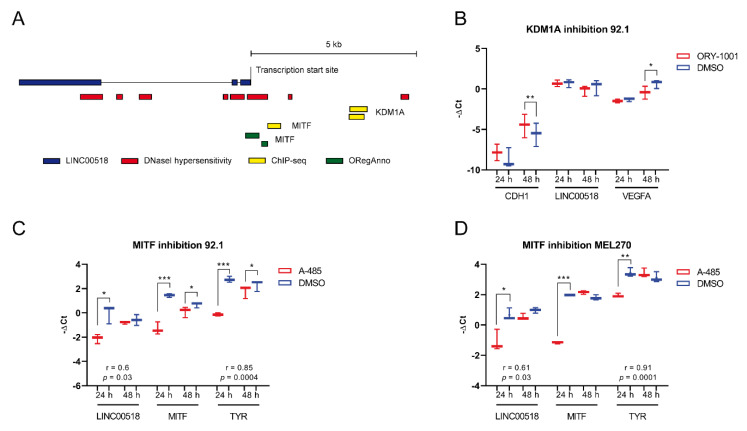
Identification of TFs potentially regulating *LINC00518* expression. (**A**) Experimentally validated transcription factor binding sites (TFBSs) upstream of the transcription start site of *LINC00518* locus. (**B**) Expression analysis after *KDM1A* inhibition achieved by ORY-1001. (**C**,**D**) Expression analysis after *MITF* inhibition through A-485 treatment in 92.1 and MEL270 cells, respectively. Correlation of expression with *MITF* is shown. Each experiment was performed in three in vitro replicates. *: *p*-value < 0.05; **: *p*-value < 0.005; ***: *p*-value < 0.0005.

**Table 1 cancers-12-03867-t001:** Expression of miRNAs potentially sponged by *LINC00518* and their mRNA targets after siRNA transfection.

		92.1			MEL270		
miRNAs Regulating mRNA Expression	miRNA/mRNA	24 h	48 h	Correlation vs. *LINC00518*	24 h	48 h	Correlation vs. *LINC00518*
	miR-143-3p	−1.08 (0.55)	1.71 (0.61)	−0.2 (0.53)	1.58 (0.51)	−1.63 (0.12)	−0.35 (0.26)
	miR-145-5p	−1.19 (0.47)	1.38 (0.46)	−0.49 (0.1)	1.98 (0.65)	1.08 (0.87)	0.54 (0.06)
	miR-199a-5p	1.23 (0.99)	1.46 (0.08)	−0.5 (0.09)	1.13 (0.53)	1.07 (0.87)	−0.16 (0.61)
	miR-212-5p	1 (0.63)	1.33 (0.86)	−0.35 (0.26)	1.6 (0.99)	−1.09 (0.51)	0.47 (0.12)
	miR-3191-3p	1.94 (0.96)	2.5 (0.08)	0.11 (0.73)	1.84 (0.33)	1.21 (0.68)	0.35 (0.26)
	miR-497-5p	−1.12 (0.48)	1.65 (0.48)	−0.48 (0.11)	1.22 (0.86)	1.19 (0.72)	0.44 (0.15)
miR-212-5p, miR-3191-3p	*AHCYL2*	−1.22 (0.18)	1.82 (0.16)	−0.38 (0.22)	1 (0.98)	1.19 (0.21)	**0.65 (0.02)**
miR-199a-5p	*CRTAP*	−1.23 (0.11)	1.11 (0.37)	0.51 (0.09)	−1.05 (0.75)	1.27 (0.13)	**0.64 (0.02)**
miR-3191-3p	*ENTPD1*	1.04 (0.96)	1.59 (0.18)	0.52 (0.08)	−1.02 (0.81)	1.03 (0.82)	**0.74 (0.005)**
miR-145-5p	*F11R*	1.33 (0.24)	1.08 (0.12)	**−0.7 (0.01)**	1.04 (0.67)	1.07 (0.54)	**0.59 (0.04)**
miR-145-5p, miR-212-5p	*IFFO2*	1.01 (0.99)	1.42 (0.26)	0.13 (0.68)	1.01 (0.87)	1.2 (0.29)	**0.73 (0.007)**
miR-145-5p, miR-3191-3p	*IP6K1*	1.13 (0.62)	2.41 (0.25)	−0.22 (0.49)	1.15 (0.61)	1.05 (0.71)	**0.74 (0.005)**
miR-3191-3p	*KCTD15*	1.02 (0.87)	−1.15 (0.1)	0.22 (0.49)	1.05 (0.72)	1.05 (0.93)	0.18 (0.57)
miR-143-3p	*KLF8*	−1.68 (0.07)	−1.12 (0.26)	0.48 (0.11)	**−1.39 (0.04)**	1.28 (0.46)	0.15 (0.64)
miR-497-5p	*LINGO2*	−1.68 (0.09)	−1.57 (0.15)	**0.59 (0.04)**	**−1.47 (0.04)**	−1.14 (0.18)	0.42 (0.17)
miR-143-3p	*NFIA*	−1.47 (0.13)	1.56 (0.25)	**0.58 (0.04)**	**−1.48 (0.04)**	1.12 (0.48)	0.34 (0.27)
miR-145-5p, miR-497-5p	*OTUD7B*	**−1.4 (0.03)**	1.26 (0.54)	0.45 (0.14)	**−1.23 (0.04)**	1 (0.59)	**0.75 (0.004)**
miR-3191-3p	*RAB43*	−1.52 (0.11)	1.27 (0.43)	0.43 (0.16)	−1.06 (0.65)	−1.11 (0.75)	**0.72 (0.008)**
miR-212-5p	*RABGAP1L*	1 (0.76)	1.55 (0.06)	−0.29 (0.36)	1.24 (0.12)	1.13 (0.25)	0.31 (0.32)
miR-199a-5p, miR-212-5p	*SEC22C*	**−1.31 (0.001)**	1.36 (0.16)	−0.11 (0.73)	**−1.11 (0.03)**	**−1.07 (0.01)**	**0.71 (0.009)**
miR-497-5p	*SEMA6A*	1.38 (0.17)	1.38 (0.34)	**−0.58 (0.04)**	−1.03 (0.99)	**−1.06 (0.04)**	0.41 (0.18)
miR-3191-3p	*VAMP3*	**−1.37 (0.02)**	1.31 (0.25)	**0.58 (0.04)**	−1.05 (0.26)	1.36 (0.18)	**0.58 (0.04)**
miR-199a-5p	*ZBTB20*	−1.03 (0.75)	−1.1 (0.17)	−0.31 (0.32)	1.22 (0.19)	1.21 (0.35)	0.04 (0.9)

For each miRNA/mRNA target, average fold change and *p*-value (between brackets) are reported for each time point. R-values representing correlation of expression with *LINC00518* and the associated *p*-values (between brackets) are shown. For each mRNA, miRNAs regulating its expression are reported. Significant results are highlighted in bold. Each experiment was performed in three in vitro replicates.

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
