# Peer review of "LncRNA LINC00518 Acts as an Oncogene in Uveal Melanoma by Regulating an RNA-Based Network"

_cancers, 2020, doi:10.3390/cancers12123867_

Round 1

Reviewer 1 Report

The paper is about an interesting topic, and results result robust and experiments well-designed. Authors address the putative oncogenic role of a new lncRNA in Uveal melanoma, providing evidence of its involvement in processes related to tumorigenesis, suggesting some interactors in consideration of ceRNA hypothesis and transcriptional regulators. The paper is well-written and explained. This analysis is mainly based on in silico-data and correlation analysis, thus further experiments should improve the quality and strengthen of the conclusions.

Particularly, authors should perform few targeted experiments and consider some modifications in presentation of the results:

To further support the proposed “ceRNA network” addressing the ability of LINC00518 to modulate selected mRNAs by competions with miRNAs, authors should evaluate the ability of selected miRNA to directly regulate mRNA targets in used cell lines (so in the exact cellular context) by miRNA mimics, then evaluating the expression of selected targets. This could mimick the effects of LINC00518 silencing.

To definitively demonstrate that LINC00518 is targeted by MITF (and not KDM1A) in used cell model, authors should perform by gene silencing of MITF (better if also over-expression) and preferentially also for KDM1A, and then evaluating LINC00518 expression. The only use of inhibitor (with could produce multiple and indirect effects, also potentially masking expression variations) is an useful approach to combine to gene modulation but alone is not sufficient for definitively demonstrating or excluding putative direct regulation.  Furthermore, inhibitor treatments should be performed on both cell lines, as also gene silencing (and eventual over-expression). 

The authors conclude that “Unfortunately, HGF failed to induce EMT in 159 MEL270 cells, despite the increased concentration used to treat the cells (Table 1)”. The bona fide of experiment should be confirmed by migration assay.

Authors should be better state the selection criteria of mRNA chosen as targets of LINC00158, since they state that "For mRNAs regulated by LINC00518 silencing, expression and correlation analyses were also 230 performed on UM biopsies" refferring to LINGO2, NFIA, OTUD7B, SEC22C, and VAMP3. But as they declare "overall, LINGO2, NFIA, OTUD7B, SEC22C, and VAMP3 217 downregulation after LINC00158 silencing was reported for both cell lines" but in 92.1 cells they state that "a significant positive correlation of expression with LINC00518 212 was observed for LINGO2, NFIA and VAMP3" so not for OTUD7B, SEC22C".

Overall, when cite gene name, so about gene expression data, gene name have to be written in italic.

Expression data are showed as DCt, rather than analysed by the most used methods DDCt. Can the authors explain the reason? DDCt methods allow a more immediate representation of data, usually indicating the value in reference sample as 1 and showing the expression of testing samples as fold variation.

Similarly to previous point, if no specific motivation justify data representation, also data on migration and viability assays should be represented as relative compared to control cells (posing for example the viability or migration in control as 100% and showing the variation in percentage). Furthermore, I suppose that in the Figure 3 the * indicating significance are referred to the comparison between NC and silenced cells, so they have to indicate between the two bars and not on bar of silenced cells.

Line 452, authors should indicate the eventual number of healthy samples in the UM dataset of The Cancer Genome Atlas (TCGA), specifying if are healthy counterparts or if only tumor samples are available. This is relevant to clarify if the showed correlations have been performed only in tumor condition or also considering healthy tissues.

Authors should better explain the selection criteria for LINC518, given that in the paragraph 2.2 the correlation with BAP1 (similar to that with MET, so “not linear”) is near to the significance, and however also LINC634 shows correlations with MET and BAP-1.

In Figure 2c, the values are comparable to that obtained in DMSO, what is the control? Maybe an internal control for each time point? But it is “strange” that DMSO have similar values, please specific why and the significance to that variation (compared to what control) is referred. Also in this case, a relative representation of data (for example showing the control as 1) should be help to give more clarity.

Please add “cells” to the title of Figure 2.

Hours indicated as h should be near 24, 48 etc without space.

Please cite literature evidences that IL6 and HGF treatment are sufficient as EMT stimuli.

Please modify the sentence “In contrast, when lncRNAs sequester miRNAs, their mRNA targets are not degraded and can be translated” in “In contrast, when lncRNAs sequester miRNAs, the degradation and/or translation of mRNA targets is reduced”

It is more appropriate do not define as “direct” the involvement of LINC00518 in cancer-related processes, so I suggest to remove “direct” in this sentence: “The detection of LINC00518 dysregulation associated with several functional in vitro assays allowed us to investigate its ceRNA regulatory network and shed light on its direct involvement in cancer-related processes.”

Similarly to previous point, modify the sentence “The increased expression of LINC00518 observed in UM tissue suggested an oncogene role for 189 this lncRNA” at line 189, in “The increased expression of LINC00518 observed in UM tissue suggested a putative oncogenic activity for this lncRNA”

Please specify that Mel270 cells and 92.1 cells are both uveal melamoa cells

In the discussion, authors should discuss a possible explanation of different effect of hgf in the used cell lines, if any.

Significant data showed in table 1,2,3, 4 should be showed as graph and move complete tables in supplementary, especially for Table 3. Indeed, data about mRNAs in Table 3 as graph could result more immediate. A bargraph with variations of mRNA following LINC silencing (possibly as Fold changes) could be relevance to this result that in my opinion, is a crucial point.

Authors should correct the sentences where they declare that correlation data demonstrate LINC00518 to regulate selected mRNAs, since they only support/suggest it. So, also in consideration to previous point and data in Table 3, authors should be consider that, clearly, correlation analysis do not indicate causality, so only support the possible ability of LINC00518 to regulate mRNAs. The major experiment addressing the putative causality is the silencing of  LINC00518 which induces mRNA downregulation (also for this reason this result should be clearly showed in a bargraph, as indicated in previous point).

Please, explain the motivation of the investigation for expression variation of six miRNAs, what is the hypothesis about this point?

In table 3 caption, authors declare that significant results are highlighted in bold, but bold is not visible.

Authors should correct multiple typos, such as:

- Line 25-99-409 close parenthesis

- Line 49-74 remove the additional parenthesis

- Line 142 adjust in capital letter

Authors should cite in Introduction very recent reviews about the relevance of the LincRNA in tumorigenesis to strengthen the relevance of their work and obtained results, useful especially for not fully addicted readers. Some suggestions could be:

-Aprile, M.; Katopodi, V.; Leucci, E.; Costa, V. LncRNAs in Cancer: From garbage to Junk. Cancers 2020, 12, 3220.

-Zhang, X.-Z.; Liu, H.; Chen, S.-R. Mechanisms of Long Non-Coding RNAs in Cancers and Their Dynamic Regulations. Cancers 2020, 12, 1245.

-Carlevaro-Fita, J., Lanzós, A., Feuerbach, L. et al. Cancer LncRNA Census reveals evidence for deep functional conservation of long noncoding RNAs in tumorigenesis. Commun Biol 3, 56 (2020). https://doi.org/10.1038/s42003-019-0741-7

-Jiang MC, Ni JJ, Cui WY, Wang BY, Zhuo W. Emerging roles of lncRNA in cancer and therapeutic opportunities. Am J Cancer Res. 2019;9(7):1354-1366. Published 2019 Jul 1.

Reviewer 2 Report

In this manuscript, the authors show the role of the lncRNA LINC00518 as an oncogene in UM. While initially identified through the analysis of TCGA data, they validated the expression in an independent set of UM biopsies. While this lncRNA does correlate with BAP1 and MET expression its expression is not correlated with any clinicopathological attributes. Regardless of this, they show that it has a direct role in EMT and response to hypoxia, cell proliferation and migration. They hypothesize that this gene is functioning in a network interacting with both miRNAs and mRNAs as both a miRNA sponge and protector. They further show potential mechanisms of its regulation of expression via MITF and other transcription factors. Overall, they lay out a compelling story for the role of this lncRNA in UM, and lncRNAs as a whole. The work is well done, experiments are well controlled. A few comments:

There was no association of the lncRNA to the clinicopathological attributes in the biopsy samples, was this also true for the TCGA data?

Potential miRNA binding sites were predicted using 2 computational programs. It is has been well established that many of these programs have a high false-positive rate. It would be useful to experimentally validate the putative binding sites. I would be very surprised if all of those sites successfully validated.

In figure 6, what is the scale in the x-axis? Are those the positions along the 3UTR of each of the genes?

The lncRNA:mRNA interactions are not clear to me. Is the suggestion here that the lncRNA is binding to the complete span of the 3UTR or just specific positions (based upon the free energy)? Similarly, is there a potential “cut-off” as to what would be considered as a significant free energy value?

In Figure 7, is there a reason that KDM1A inhibition was not done in MEL270 cells?

Reviewer 3 Report

Authors presented the results related to the lncRNA LINC00518 in uveal melanoma. Manuscript relates to very important issue, but some corrections are necessary.

  1. LINC00518 has been studied in melanomas (cutaneous), but Authors did not discuss their results with other published data properly, for example:
    -Melanoma Res. 2018 Oct;28(5):478-482. doi: 10.1097/CMR.0000000000000478.
    -Skin Therapy Lett. 2018 Sep;23(5):1-4.
    -J Am Acad Dermatol. 2017 Jan;76(1):114-120.e2. doi: 10.1016/j.jaad.2016.07.038. Epub 2016 Oct 1.
    -JAMA Dermatol. 2017 Jul 1;153(7):675-680. doi: 10.1001/jamadermatol.2017.0473.
  2. Authors stated that "Selection of lncRNAs potentially involved in UM molecular processes was performed by using transcriptomic data available at TCGA [38]" - but this reference does not fit to the information presented in the sentence.
  3. What was the tissue size for RNA isolation? Authors stated that it was 10um slice, but there is not information about the weight or diameter of the slice.
  4. Why Subcellular fractionation was performed only for 92.1 cell line? Why different experiments in vitro were performed on different cell lines? Experiments on one cell line in not acceptable.
  5. Description of the in vitro methods is superficial and insufficient. For example:
    -how hypoxia was induced
    -how cells were for MAPK inhibition, Cell cycle block and activation , EMT induction and hypoxia treated with LINC00518.
  6. What does mean "six biological replicates"?
  7. The number of samples for each experiment or number of experiments in vitro should be included in the legend figures.
  8. English correction should be done. There are some typos.

Round 2

Reviewer 1 Report

The reviewer thanks authors for answers, and approves performed modifications. 

Only few suggestions/considerations about specific points are indicated below (in bold):

1. To further support the proposed “ceRNA network” addressing the ability of LINC00518 to modulate selected mRNAs by competions with miRNAs, authors should evaluate the ability of selected miRNA to directly regulate mRNA targets in used cell lines (so in the exact cellular context) by miRNA mimics, then evaluating the expression of selected targets. This could mimick the effects of LINC00518 silencing.

This approach could obviously confirm the causal relationship between miRNAs and mRNAs. As described in Methods, mRNAs selection was carried out by considering three different parameters compatible with the miRNA sponge model: 1) positive correlation with LINC00518 in UM samples, 2) negative correlation with the selected miRNAs in UM samples, 3) presence of multiple binding sites for at least one of the selected miRNAs. Although we cannot definitively state that these mRNAs are effectively regulated by the selected miRNAs within cells, we have strong suggestions that this interaction may occur in UM cells. Also, we did not perform miRNA mimic transfection in this paper because we only proposed a potential miRNA sponge model for LINC00518 by showing its systemic effects on multiple miRNAs and mRNAs, without focusing on specific LINC00518:miRNA interactions, whichwe plan to analyze in further studies.

I consider that, in line with the quality of this journal, this aspect should be strengthen, thus could be at least possible to find literature data or analyze public datasets of overexpression/inhibition for selected miRNA (if any) to assess the effects of this miRNA on biological processes related to LINC00518 silencing?. I also suggest  a clear declaration in abstract or conclusion similarly to that authors indicate in the answer: “we only proposed a potential miRNA sponge model for LINC00518 by showing its systemic effects on multiple miRNAs and mRNAs, without focusing on specific LINC00518:miRNA interactions, whichwe plan to analyze in further studies”

  1. To definitively demonstrate that LINC00518 is targeted by MITF (and not KDM1A) in used cell model, authors should perform by gene silencing ofMITF (better if also over-expression) and preferentially also for KDM1A, and then evaluating LINC00518 expression. The only use of inhibitor(with could produce multiple and indirect effects, also potentially masking expression variations) is an useful approach to combine to gene modulation but alone is not sufficientfor definitively demonstrating or excluding putative direct regulation. Furthermore, inhibitor treatments should be performed on both cell lines, as also gene silencing (and eventual over-expression).

Inhibition of KDM1A was first performed in 92.1 cells and showed no significant results in LINC00518 deregulation, thus suggesting that this TF may not be involved in expression regulation of this lncRNA. Given the non-significant result, we did not perform this experiment in MEL270. Indeed, we planned to consider as significant only results confirmed in two different cell lines. Results not significant in the first cell line where not tested in the second, since they would not have been considered as reliable even if statistically significant. We did not perform siRNA silencing of MITF, as we did not perform siRNA silencing of MEK1/2 to inhibit MAPK pathway. For both experiments, we have used commercial inhibitor which have been validated and are currently used for their purpose. From a functional point of view, the effects of siRNAs and chemical inhibitors are almost the same, with the difference that siRNA is easier to perform but it can also have off-target effects, causing cellular toxicity and innate immune response(Recognizing and avoiding siRNA off target effects for target identification and therapeutic application, Nat Rev Drug Discov. 2010). Also, as demonstrated by reduction of both MITF and TYR mRNAs, MITF inhibition was successfully achieved, suggesting that the reduced expression of LINC00518 is due to reduced activity of the TF. Also, we only suggest that MITF is responsible for LINC00518 increase expression in UM. In our opinion, siRNA silencing in this case is not mandatory to suggest a regulatory mechanism operated by the TF.

Please specify this aspect in methods

  1. The authors conclude that “Unfortunately, HGF failed to induce EMT in 159 MEL270 cells, despite the increased concentration used to treat the cells (Table 1)”. The bona fide of experiment should be confirmed by migration assay.

Evaluation of efficacy of EMT induction for both IL6 and HGF was performed by analyzing mRNA levels of EMT markers, namely CDH1, VIM and ZEB2;a similar approach was followed by other authors (Miao et al., Int J Oncol. 2014;45:165-76). Since there is a lot of evidence showing IL6 and HGF capability to induce EMT, we did not perform migration assay to confirm EMT induction.Evaluation of efficiency by EMT marker expression, which show molecular changes related to EMT, was sufficient for our scope, that is investigation on molecular modulation of LINC00518.

Since drugs can be similarly have unwanted or indirect effects by targeting of multiple pathways, please indicate if literature supports that these drugs have a good target specificity. However, other journals with lower quality/IF higher, as Int J Oncol, obviously can require a minor number of data/experiments, but this does not demonstrate that this approach is complete and cannot be improved.

  1. Authors should be better state the selection criteria of mRNA chosen as targets of LINC00158, since they state that "For mRNAs regulated by LINC00518 silencing, expression and correlation analyses were also 230 performed on UM biopsies" refferring toLINGO2, NFIA, OTUD7B, SEC22C, and VAMP3. But as they declare"overall, LINGO2, NFIA, OTUD7B, SEC22C, and VAMP3 217 downregulation after LINC00158 silencing was reported for both cell lines" butin92.1 cells they state that"a significant positive correlation of expression with LINC00518 212 was observed for LINGO2, NFIA and VAMP3" so not forOTUD7B, SEC22C".

We thank the reviewer for this question that allows us to better explain this point. Selection of mRNAs considered as targets of LINC00518 was carried out according to two evidence: i) obviously, decreased expression of the mRNA after LINC00518 transient silencing, suggesting a regulation operated by the lncRNA; ii) the conservation of positive correlation of expression with the LINC00518 in cells where the lncRNA expression is transiently reduced, which suggests an involvement of LINC00518 in the regulation of mRNA expression.Moreover, mRNAs were considered as LINC00518 targets only if significant results in expression modulation or correlation were observed in both cell lines. Indeed, despite some mRNAs showed a non-significant downregulation, reduced levels of these mRNAs were observed with low p-values which did not reach statistical significance. This downregulation trend, together with positive correlation of expression, led us to consider these mRNAs as potentially regulated by LINC00518. Of course, our data does not prove definitively whether this regulation is direct or indirect. We proposed two hypothesis explaining both a direct regulation (miRNA protector model) and an indirect regulation (miRNA sponge model).

Please specify this aspect in methods

  1. Expression data are showed as DCt, rather than analysed by the most used methods DDCt. Can the authors explain the reason? DDCt methods allow a more immediate representation of data, usually indicating the value in reference sample as 1 and showing the expression of testing samples as fold variation.

We have always shown data in box-plots as DCts since these represent the primary normalized expression data of a sample (without any other mathematical elaboration). Especially when showing patient data, where high variability is very common, DCts allow to have a precise vision of expression data for both affected and unaffected individuals.Also, we prefer to show DCts because our analyses include paired samples. Plotting DDCts would have led to the loss of standard deviation for control samples, as all of them would have assumed a value equal to 0.

I understand authors’ considerations. However, if possible I should appreciate if they could confirm me (and eventually indicate them) that FC values obtained using DDCt method are substantial and statistically significant. To the best of my knowledge, DDCT method is the standard method used for relative qPCR assays, and DCt can amplify some variations. However, usually control samples are assumed to 1 and standard deviation of control can be recovered analyzing them vs mean DCts of controls.

  1. Authors should cite in Introduction very recent reviews about the relevance of the LincRNA in tumorigenesis to strengthen the relevance of their work and obtained results, useful especially for not fully addicted readers. Some suggestions could be:

-Aprile, M.; Katopodi, V.; Leucci, E.; Costa, V. LncRNAs in Cancer: From garbage to Junk. Cancers 2020, 12, 3220.

-Zhang, X.-Z.; Liu, H.; Chen, S.-R. Mechanisms of Long Non-Coding RNAs in Cancers and Their Dynamic Regulations. Cancers 2020, 12, 1245.

-Carlevaro-Fita, J., Lanzós, A., Feuerbach, L. et al. Cancer LncRNA Census reveals evidence for deep functional conservation of long noncoding RNAs in tumorigenesis. CommunBiol 3, 56 (2020). https://doi.org/10.1038/s42003-019-0741-7

-Jiang MC, Ni JJ, Cui WY, Wang BY, Zhuo W. Emerging roles of lncRNA in cancer and therapeutic opportunities. Am J Cancer Res. 2019;9(7):1354-1366. Published 2019 Jul 1.

If authors agree and consider this appropriate, the sentence could be referred specifically to lncRNAs rather than ncRNAs, also highlithing the therapeutic opportunities of this class of ncRNAs.

Reviewer 2 Report

The authors have sufficiently addressed my questions.

Author Response

We thank the reviewer for his/her useful suggestions.

Reviewer 3 Report

Authors adequately addressed only some of my comments.

-Authors should include responses to my comments to their manuscript, for example “Each 10um slice has a weight of approximatively 0.01 g; the diameter is comprised between 1.5 and 2.5 cm.”
-Authors cited the articles they mentioned in the response letter and articles I suggested, but they did not discuss properly their results with the results of others authors. Is should be corrected.
Similarly, Authors did not discuss their results in relation to starvation.
-Citing the TCGA is nor proper and referring to other authors is also impropriate. Authors should cite TCGA data according to rules indicated by TCGA (on TCGA website).
-In addition, Authors should provide the introduction about melanin since melanin content is very important aspect of melanoma biology (see and cite: Physiol Rev. 2004. PMID: 15383650 Review).

-Melanin content in analyzed cells should be also discussed and analyzed, since active melanogenesis:
A) contributes to melanoma resistance to chemo-, radio- and immunotherapy (see and cite:
--Int J Mol Sci. 2018 Apr 1;19(4):1048. doi: 10.3390/ijms19041048
--Int J Cancer . 2009 Mar 15;124(6):1470-7.
doi: 10.1002/ijc.24005)
B) and melanoma pigmentation affects the survival of both cutaneous and uveal melanoma patients (see and cite:
--Eye (Lond). 2015 Aug; 29(8): 1027–1035. Published online 2015 Aug 7. doi: 10.1038/eye.2015.51).
